# Changes in systems thinking and health equity considerations across four communities participating in Catalyzing Communities

Travis R. Moore[1,2]*, Larissa Calancie[1], Erin Hennessy[1], Julie Appel[1], Christina D. Economos[1]

1 ChildObesity180, Friedman School of Nutrition Science and Policy, Tufts University, Boston, MA, United States of America, 2 Department of Community Health, Tufts University, Medford, MA, United States of America

* travis.moore@tufts.edu

**Data Availability Statement:** All participant interviews (all data) are available at https://osf.io/vcdyg/ and via DOI 10.17605/OSF.IO/VCDYG.

## Abstract

Childhood obesity is a persistent public health concern, and community-based interventions have become crucial for addressing it by engaging local communities and implementing comprehensive evidence-based strategies. The Catalyzing Communities intervention takes a "whole-of-community" approach to involve leaders from diverse sectors in thinking systematically about child healthy weights and implementing evidence-based solutions. Using systems thinking and the Getting to Equity framework to guide interview analysis, this study examines changes in participants' use of systems thinking concepts and health equity in 43 participants across four U.S. communities involved in the Catalyzing Communities intervention. Our findings reveal significant shifts in systems thinking concepts, as participants develop a deeper understanding of childhood obesity as a complex adaptive system, and system insights, as participants increasingly recognize the interconnections and leverage points within the system driving childhood obesity. Participants also experienced increases in health equity thinking and action, particularly when discussing social and structural determinants of health, commitment to targeted actions, and a focus on addressing barriers and enhancing resources. The intersection between systems insights and health equity action, such as explaining leverage points and interventions to reduce deterrents to health behaviors, suggests the need for systems thinking activities to be integrated into health equity planning. Future research is needed to develop measures to connect systems thinking concepts to health equity, and the impact of these to community-level policy, systems, and environmental changes in public health.

## Introduction

Childhood obesity is a pressing concern in contemporary public health, characterized by its decades long persistent prevalence, recent rising incidence, and significant long-term health

**Funding:** C.E. received funding that supported this study from the JPB Foundation (PR0580). The JPB Foundation did not play a role in the study design, data collection and analysis, decision to publish, or the preparation of the manuscript. https://www.jpbfoundation.org.

**Competing interests:** The authors have declared that no competing interests exist.

and societal consequences [1]. Exacerbated by the COVID-19 pandemic [2], childhood obesity has prompted extensive research and intervention efforts aimed at curbing its impact [3]. One approach in this endeavor is the implementation of community-based interventions that engage local communities in implementing multifaceted strategies to address childhood obesity comprehensively [4]. By leveraging the strengths, resources, and knowledge present within these communities, such interventions aim to create sustainable changes in child health behaviors, environments, and the policies that influence them [5].

The "Catalyzing Communities" (CC) intervention represents an important initiative within this landscape [6, 7]. Recognizing the multi-level complexity of childhood obesity and the need for holistic community engagement, this intervention adopts a "whole-of-community"approach and builds on over 20 years of our team's research [8–10]. It seeks to forge deep relationships among a core group of leaders from various sectors (e.g., healthcare, early education, philanthropy) to think systemically about a topic related to child healthy weights. The intervention applies community-based system dynamics [11], an approach popularized by Peter Hovmand that integrates systems thinking and participatory methods to engage communities in understanding and addressing complex issues [11]. The participatory method deployed in CC is group model building (GMB), a community-driven approach designed to leverage the collective knowledge and insights of participants to create models (in this case, causal loop diagrams) that accurately reflect the perspectives on and complexities of promoting health in children [11]. Ongoing support and capacity-building are provided to guide the selection of actions to implement and ensure the sustainability of positive changes. Interventions that use similar approaches have been shown to increase systems thinking [12, 13], and the Catalyzing Communities intervention has recently been cited as a high quality application of community-based system dynamics [13].

In this context, CC functions as a catalyst for transformation, fostering collaboration and encouraging a paradigm shift in how communities approach child health. Part of understanding its impact is understanding how participants' systems thinking concepts, understanding of health equity, and perceptions of community transformation are associated. Such changes can have far-reaching implications, not only for child health but also for the broader field of community-based interventions and public health [12]. Therefore, this study explores the shifts in participants' systems thinking concepts, system insights, and health equity thinking and action across groups of stakeholders working in four U.S. communities engaged in the CC intervention.

The research team selected the four evaluation questions to examine changes in systems thinking concepts, system insights, and health equity thinking and action among participants before and after the Catalyzing Communities (CC) intervention. These questions were used as a framework to develop a codebook for analyzing participant interviews. The four questions are:

- To what extent do CC participants use systems thinking concepts to discuss childhood obesity and healthy child weights in their communities before and after participating in the intervention?

- What levels of systems insights do participants display before and after the CC Intervention?

- How do CC participants describe their thinking and action related to health equity before and after the intervention?

- How are systems insights linked to health equity thinking and action before and after the CC Intervention?

Health equity played a crucial role in the research by examining how participants' understanding and actions related to health equity evolved throughout the intervention. Although the concept of health equity was not explicitly integrated into the development of interview questions, it emerged as a significant theme during analysis and became an integral part of the research focus. The evaluation questions were designed to capture any changes in participants' health equity thinking and action, with the aim of examining how systems thinking concepts and system insights might be linked to health equity thinking and action.

Each of these questions were examined across four CC intervention sites. Participants were interviewed at the beginning of the intervention (after participants are convened) and approximately one year into a multi-year intervention. We wanted to assess the effect of participating in the first phase of the intervention, which is focused on building relationships and understanding systems that create important community health trends, on participants' use of systems thinking concepts. By also coding and analyzing these interviews for systems insights and health equity thinking and action, we begin to link whether and how participants' system insights about the drivers of child health inequities are related to health equity thinking and action in their local communities. In doing so, we contribute to the growing research and practice evidence that suggests systems thinking is a salient component of working toward mitigating child health inequities. This research has potential implications for researchers and practitioners developing or engaged in whole-of-community interventions that aim to promote child healthy weights.

## Background

### Systems thinking

Systems thinking in CC broadly refers to understanding and analyzing complex phenomena as interconnected and interdependent components of a whole. For CC participants, this means adopting a holistic perspective when addressing local issues related to promoting healthy weights for children. This approach involves recognizing the broader system of interrelated components, identifying system feedback and structure, and understanding the dynamic behavior and leverage points within the system.

Though there is no universally agreed-upon definition of systems thinking, several authors informed how we define and operationalize the concepts of systems thinking in CC. Donella Meadows' "Thinking in Systems" provides a foundational framework for this approach [14]. Meadows defines systems thinking as a discipline for seeing wholes, a framework for seeing interrelationships rather than things, and for seeing patterns of change rather than static snapshots. It involves understanding elements such as feedback loops, stocks, and flows within systems and how these elements interact to produce the behavior of the system over time. Key components include recognizing interconnections, understanding complex cause-and-effect relationships, and identifying leverage points where interventions can create significant positive changes. Further, Arnold and Wade (2015) reviewed relevant literature (e.g., Hopper & Stave, 2008 [15]; Plate & Monroe, 2014 [16]) to distill eight essential elements in the context of understanding the dynamics of systems: recognizing interconnections, identifying and understanding feedback, understanding system structure, differentiating types of stocks, flows, and variables, identifying and understanding non-linear relationships, understanding dynamic behavior, reducing complexity by modeling systems conceptually, and understanding systems at different scales [17]. We integrated Meadows' and Arnold and Wade's elements of systems thinking to build our codebook of systems thinking concepts.

We also adapted Hovmand's surface, mid, and deep-level system insights to situate systems thinking concepts along a spectrum of more actionable understandings or discoveries derived

from analyzing a system [11]. Surface-level insights acknowledge the existence of a broader system of interrelated components, recognizing that issues like child health are influenced by various factors such as nutrition, physical activity, socioeconomic status, and community resources. Mid-level insights delve deeper by identifying feedback loops and system structure, analyzing how different elements interact and understanding the cause-and-effect relationships that can amplify or stabilize system behavior, such as how community interventions can support or hinder efforts to promote healthy weights among children. Finally, deep-level insights explain dynamic behavior over time and identify leverage points for intervention. At this level, participants can pinpoint areas where strategic changes, such as policy implementations, can lead to significant positive impacts on the system [11]. This spectrum of system insights informed our system insights codes.

## Health equity thinking and action

Systems thinking serves as a valuable tool for understanding and addressing the complexities of health disparities for the purpose of promoting health equity by encouraging a holistic perspective and identifying root causes, the interconnectedness of factors, and feedback loops. Health equity is a guiding principle that stems from a recognition of the unjust and avoidable disparities in health outcomes experienced by different population groups [18]. Health equity acknowledges that the distribution of health and well-being should be just and fair, irrespective of race, ethnicity, socioeconomic status, or other social determinants. Health equity has become a foundational principle in public health [19, 20], guiding the design and implementation of interventions that prioritize underserved communities and seek to dismantle the structural barriers that perpetuate health inequalities.

The principle of health equity guides interventions, specifically those that are community-based, to prioritize underserved communities and seek to dismantle structural barriers to health equity [21]. Community-based interventions are increasingly using community-based system dynamics and GMB to promote health equity in childhood obesity prevention by engaging communities to address obesity's root causes. In rural Victoria, Australia, the Whole of Systems Trial of Prevention Strategies for Childhood Obesity (WHO STOPS) used GMB to create causal loop diagrams for community-led planning and interventions [22]. Similarly, the Milwaukee Childhood Obesity Prevention Project used group model building to engage stakeholders in policy and environmental changes [21]. The GenR8 Change initiative in Victoria developed a causal loop diagram to prioritize actions like creating sugar-free zones and increasing physical activity [23]. In Greenville County, South Carolina, a stakeholder-driven intervention used community-based system dynamics and GMB to address food insecurity and support systemic advocacy [24].

The current study builds upon these studies by exploring how systems thinking concepts can be linked to promoting health equity and actionable outcomes, examining the ways in which the integration of community insights and systemic approaches not only identifies underlying causes of health disparities but also facilitates tangible changes in policies and practices that advance equity. To link systems thinking concepts to health equity, we used Dr. Shiriki Kumanyika's Getting to Equity framework, that provides a valuable perspective on how to address health equity within the context of childhood obesity prevention [25, 26]. Separated into four synergistic categories, the framework integrates emerging understandings and approaches in the broader field of health equity practice and research to translate intention to achieve equity into health equity action. Firstly, "Increasing Healthy Options" involves enhancing access to nutritious food by, for example, improving the locations and marketing practices of supermarkets, establishing food standards in various settings, and making

neighborhoods more conducive to physical activity. Secondly, "Reducing Deterrents" focuses on mitigating factors that discourage healthier choices, such as countering the disproportionate marketing of unhealthy foods to marginalized communities and implementing policies like sugary beverage taxes. Thirdly, "Improving Social and Economic Resources" aims to leverage government and charitable programs to address hunger, poverty, and broader disparities while considering individual and community-focused interventions. Lastly, "Build on Community Capacity" emphasizes Policy, System, and Environmental changes driven by community engagement and capacity-building, spanning sectors like housing, education, and economic development, with an emphasis on empowering communities to take control of their health outcomes while promoting health equity in childhood obesity prevention efforts.

In this study, the Getting to Equity framework was applied post hoc to categorize the thinking and possible actions taken by intervention participants to address the root causes of childhood obesity. We used this framework to systematically analyze and categorize the various potential strategies and interventions described by interview participants. This approach allowed for a clearer understanding of how different actions aligned with the four Getting to Equity categories: enhancing access to healthy options, reducing deterrents to healthy living, improving social and economic resources, and building community capacity. The post hoc application of this framework provided a structured lens through which to interpret the data, highlighting how systems thinking concepts, system insights, and community engagement efforts translated into practical actions aimed at promoting health equity in childhood obesity prevention.

## Whole-of-community interventions

The paradigm of health equity helps shape whole-of-community interventions, which emerged as a response to the limitations of siloed and top-down approaches to public health [27]. These interventions actively engage communities in identifying and addressing their unique health challenges and disparities at multiple levels and multiple settings, fostering collaboration among diverse stakeholders, and empowering local leaders. This historical shift towards community-driven initiatives has been instrumental in achieving sustainable improvements in public health outcomes, as it recognizes the importance of local context, community expertise, and community engagement and ownership in shaping health interventions. Whole-of-community approaches have become increasingly influential in public health, promoting inclusivity, community resilience, and promoting health equity [28].

Researchers have called for integrating systems thinking into multi-level approaches to tackle health inequity [29, 30]. And multi-level interventions designed to promote systems thinking among community leaders and to create ripple effects that influence changes in community policies, systems, and environments that affect child health are now being tested [6, 7, 12]. For the whole-of-community interventions that use participatory approaches like GMB, researchers report shifts in mental models and community consensus on obesity drivers, resulting in tailored interventions [22]. Successful projects, such as those in rural Australia [31] and urban Milwaukee [21], used GMB to promote systems thinking and develop causal loop diagrams that guided community-led actions. Additionally, whole-of-community initiatives like GenR8 Change show how participatory methods foster collective action and sustainable change, emphasizing the value of community involvement in public health interventions [23].

Early evidence from more targeted interventions, those that work with single agencies or evaluation sites, also suggests that specific modeling processes (e.g., GMB) can be used to identify influential feedback systems driving health inequities [32]. For example, Owen et al. (2018) highlighted the dynamic elements of successful early childhood obesity prevention initiatives

through causal loop diagrams [33], while Brennan et al. (2015) emphasized the role of systems science tools combined with GMB techniques to improve community understanding of the complexity of obesity [32].

In contrast to these studies, which primarily focused on understanding and mapping obesity determinants, the current study uniquely bridges systems thinking concepts and system insights with actionable health equity efforts, thereby complementing the work of Fraser et al. (2022), who called for more practice-based research to ensure interventions are adaptable and sustainable [34].

## The whole-of-community intervention: Catalyzing Communities

The CC intervention is a whole-of-community intervention that incorporates systems thinking methods with the goal of promoting health equity in partnership with communities. The CC Intervention is based on the Stakeholder-driven Community Diffusion (SDCD) theory [6]. The SDCD theory draws from Community Coalition Action Theory and the Community-based Participatory Research conceptual model to describe how community coalitions have the potential to be a community-led intervention aimed at community-level change, particularly as it relates to building community capacity through synergistic pooling of resources and intellectual competencies. Based on the SDCD theory, the CC intervention includes four key activities: 1) convening a multisector group of stakeholders to address child health; 2) participating in GMB activities to align the diverse perspectives of stakeholders, surface important systems insights about child health equity in the community, increase knowledge of and engagement around child health, and build new relationships; 3) prioritizing of action strategies for intervention in the system, based on customized technical assistance for research evidence use and uptake; and finally 4) allocating project seed funding towards action identified by the group and providing continued technical assistance for action implementation and seeking funding to sustain community-driven momentum. This four-part intervention process is hypothesized to result in the diffusion of knowledge and engagement in child health, as well as the use of systems thinking concepts, through social network connections, into the organizations of the stakeholders who participate in the intervention and beyond to the larger community. The resulting diffusion is hypothesized to increase the prioritization of child health initiatives among community decision-makers and organizations, which catalyzes and reinforces important policy, practice, and environmental improvements at the community level, that influence improved child health outcomes at the individual level. More detail about CC, including recruitment procedures, can be found elsewhere [6, 8, 24].

In CC, GMB is the main method believed to increase systems thinking by brining individuals together to engage in structured activities to explore the systems and dynamics driving a trend of interest [35]. GMB emerged as a tool to elicit and share information about complex systems and often to build consensus about effective actions to shift a system in order to produce more favorable trends. A review of GMB studies found that the process may be particularly effective at supporting communication and consensus within groups [36]. Seen in S1 Table, we include GMB activities in the CC intervention because it is very well suited for facilitating discussions about complex topics, like promoting healthy child weights in communities where there are health disparities, with groups of people and building alignment for action. Some GMB practitioners place additional attention on managing power dynamics during the process and empowering communities to building modeling capabilities, which, along with predefined GMB activities from Scriptapedia [37], are elements we incorporated into our GMB activities in CC [11]. There are multiple studies detailing our application of GMB in the CC intervention [6, 7, 24].

As seen in S1 Table, adapted from Calancie et al. (2022), GMB activities (e.g, selecting a reference mode, variable elicitation, graphs over time, connection circles, causal loop diagrams, and selecting action ideas) help participants identify key patterns and trends, understand the components and interrelationships within the system, visualize changes over time, and see how variables are interconnected [7]. By mapping out cause-and-effect relationships and revealing feedback loops, participants can recognize leverage points and design interventions. This structured process can enhance participants' ability to grasp complex interdependencies and develop comprehensive strategies, facilitating communication, consensus, and empowered action towards addressing systemic issues [38, 39].

## Methods

### Sample

Pre- and post-intervention interviews were conducted via phone and audio recorded at the researchers' workplace with a total of 43 pre-post-matched participants (Table 1). Participants were recruited via email from February 1, 2020, to January 25, 2021, and provided written consent. Participants came from four different communities in the U.S. and were focused on different, though related, areas of promoting child healthy weights in their local community. For interviews, we used total population sampling, a type of purposive sampling where the entire population that participates in the intervention is interviewed. No participants refused to be interviewed at pre-intervention. Fifteen participants across the four communities did not respond to the request for a post-intervention interview. Aside from the interviewer and the intervention participant, no one else was present for the interviews, which lasted approximately 30 minutes. Notes were taken on the interview protocol during interviews but not used in the current study. Transcripts were not returned to participants for comment and/or correction. This study was approved by the [redacted] Social, Behavioral, and Educational Research IRB. Each participant provided written or oral consent. The COREQ checklist (S1 Checklist) was used for reporting qualitative research [40].

The interviews were conducted by three researchers: TM (PhD, postdoctoral scholar, male, with formal training in conducting interviews and over a decade of experience), LC (PhD, assistant professor, female, with formal training in conducting interviews and over a decade of experience), and JA (MS, senior project manager, female, with over five years of experience in conducting interviews). Aside from the initial contact for recruitment, there was no relationship formed with participants prior to the study commencement. Participants were informed about the general goals of the research project and were briefed on the interviewers' reasons and interests in the research.

### Interview protocol

In the development of our interview protocol (S1 File), we drew inspiration from adult transformative learning literature, particularly the works of scholars such as Jack Mezirow [41], Mary-Jane Eisen [42], and Catherine Snyder [43]. These theorists have made significant contributions to understanding how individuals undergo transformative learning experiences, challenging and reevaluating their beliefs and perspectives. The interview questions were carefully crafted to align with the principles of transformative learning, aiming to elicit reflections and shifts in participants' perceptions. Drawing from Mezirow's emphasis on critical reflection [41], our initial set of questions delved into participants' personal beliefs. By exploring concerns about childhood health and obesity prevention in their community, causes of childhood obesity, and the relationship between healthy weight and access to nutritious food, we sought to prompt participants to critically examine their own assumptions and preconceptions.

**Table 1. Summary of community and participant characteristics.**

| Community | 1 | 2 | 4 | 5 |
|---|---|---|---|---|
| **Community characteristics (2019)[3]** | | | | |
| Population estimate | 514,213 | 46,655 | 541,482 | 594,548 |
| Land area (mi [2]) | 785.0 | 4.8 | 226.7 | 96.8 |
| Median household income (USD) | $53,739 | $48,704 | $24,102 | $25,266 |
| Foreign born (%) | 7.9 | 50.4 | 15.3 | 5.0 |
| **Community race and ethnicity (%)** | | | | |
| Hispanic or Latino (all races) | 8.8 | 57.4 | 33.6 | 19.2 |
| NH* White | 69.0 | 32.6 | 62.1 | 44.8 |
| NH Black or African American | 18.0 | 2.6 | 5.2 | 38.4 |
| NH American Indian and Alaska Native | 0.2 | 0.0 | 3.7 | 0.8 |
| NH Asian | 2.2 | 3.8 | 3.2 | 4.3 |
| NH Native Hawaiian and Other Pacific Islander | 0.1 | 0.1 | 0.2 | 0.0 |
| NH some other race | 0.1 | 0.2 | 0.1 | 0.2 |
| NH two or more races | 1.7 | 3.4 | 1.6 | 2.4 |
| **Participant characteristics[3]** | | | | |
| Coalition size (n) | 19 | 15 | 11 | 13 |
| Matched participants (n) | 14 | 11 | 8 | 10 |
| Bachelor's degree and above (%) | 92.8 | 90.9 | 87.5 | 80.0 |
| Female (%) | 92.8 | 81.8 | 75.0 | 90.0 |
| Target age of intervention work | 0–18 y | 0–18 y | 0–18 y | 0–5 y |
| Coalition Focus Area(s) [1] | Policy, practice, and environmental change; Health equity; WIC [2] participation; Nutrition security and food justice; Health equity | Increase utilization of community resources among underserved populations; increasing youth physical activity; mental health | Improve school programs to increase access to healthy foods and physical activity opportunities; increase use of state tax credits for school funding; youth mental health | Improve health status of children 0–5 by increasing resource coordination across the community; advocacy for healthy environments; |

[1]Focus areas were determined through GMB activities, including evidence review. The focus areas were used in multiple correspondence analysis as part of coalition-committee dissimilarity measures.

[2]Special Supplemental Nutrition Program for Women, Infants, and Children.

[3]American Community Survey, 2019.

[4]American Community Survey, 2020.

*NH = non-Hispanic.

Building on Eisen's focus on the socio-cultural context of transformative learning, the interview protocol was extended to gather insights into prioritized actions for childhood obesity prevention and perceived barriers in the community [42]. This approach aimed to uncover the social influences shaping participants' perspectives, acknowledging that transformative

learning occurs within a broader societal and organizational context. Snyder's work on transformative learning within organizational settings guided the design of questions related to participants' organizations [43]. Participants were prompted to assess their organization's prioritization of child healthy weight efforts, its perception of childhood obesity as a community problem, and its influence on community awareness, policies, and regulations. This aligns with the idea that transformative learning extends beyond individual cognition to encompass shifts in organizational values and practices.

The interview protocol also allowed participants to explore changes in their roles, relationships, or actions resulting from their involvement with the intervention, echoing Mezirow's emphasis on the potential for transformed perspectives to manifest in behavior and actions. Furthermore, the reflective component, where participants shared insights on the intervention process and suggested improvements for future iterations, aligns with the ongoing nature of transformative learning and the continuous evolution of one's understanding.

## Data analysis

Our data analysis was guided by content analysis and involved an iterative process of codebook development, coding using NVivo software, establishing intercoder reliability, and subsequent descriptive and correlative analyses of codes and code frequencies. We began by iteratively developing a comprehensive codebook that would serve as the foundation for our qualitative analysis. This codebook included a combination of inductive and deductive codes. Initially, we developed the codebook to focus on systems thinking concepts and system insights, with health equity thinking and action emerging inductively.

We utilized NVivo software to systematically analyze our dataset. After the first author coded a single transcript, the two independent student coders were trained in the use of the codebook and then separately applied the codes to one participant's pre and post interview. This double-coding approach allowed for the assessment of intercoder reliability and consistency in code application: the two coders independently coded the sample of participant interviews that was coded by the first author. We then calculated intercoder reliability ($k = .69$, strong level of agreement) scores using Cohen's Kappa [44]. Any discrepancies or disagreements were discussed and resolved through consensus meetings, ensuring that our coding process was consistent and reliable. This coding process captured the overarching concepts found in the interviews and that aligned with our a priori codes. The final categories and their definitions with illustrative quotes are attached as S2 Table.

The open coding process employed for participant interviews was dynamic and iterative, designed to uncover new categories and insights (i.e., themes) within the data. This process was also integral to enhancing the comprehensiveness of the existing codebook. Using NVivo, a qualitative data analysis tool [45], two student coders guided by the first author started the open coding process at the same time as applying codes from the codebook, which involved systematically reviewing the interview transcripts and identifying patterns, themes, and concepts that emerged organically from the data. The coders identified instances where the emerging categories overlapped with or complemented the existing categories in the codebook. As the open coding process continued, the coders identified additional codes related to systems thinking concepts and a new category, health equity thinking. The health equity thinking codes captured participants' perspectives, attitudes, and insights regarding health equity within the context of childhood obesity, creating the categories of social determinants of health, structural determinants of health, health disparities, and intersectionality. These new categories were integrated into the codebook as inductive categories. Guided by the first author, student coders grouped these categories into broader themes based on similarities and relationships.

We also conducted descriptive analyses to summarize and compare pre-post intervention code frequencies. This is a mixed-methods approach that combines qualitative and quantitative data to provide a more comprehensive understanding of changes or trends related to specific codes. We compared the frequency of each code between the pre-intervention and post-intervention data. This comparison can reveal changes in the prevalence or salience of specific themes or concepts, as well as point to how codes are related. The results of our descriptive and correlative analyses were visualized using frequency charts and heatmaps. Frequency charts displayed the distribution of code occurrences, highlighting the most prevalent themes and patterns. The heatmap represented the changing relationships and correlations between different categories of codes, providing a graphical representation of the interconnectedness of concepts within the dataset. Using NVivo's query function, we created pre-interview and post-interview matrices to correlate the system insights codes with the health equity thinking and health equity action codes. In other words, we asked NVivo to report when two different codes overlapped in an interview and looked at the code cooccurrence frequencies across all interviews. We exported these matrices into R, a programming software, and subtracted pre-interview code cooccurrence frequencies from post-interview code cooccurrence frequencies. This resulted in a matrix of changes in code cooccurrence frequencies. Finally, we deployed an algorithm to color each cell based on the magnitude of change from pre- to post-interview to help guide result interpretation. Participants who showed up to cross-community meetings or monthly check-in meetings were given a chance to provide feedback on the findings. Feedback was used to cross-validate how findings were discussed in this paper.

## Results

### Evolving systems thinking concepts

A significant theme that emerged from the data is participants' evolving use of systems thinking concepts regarding child healthy weights. This theme highlights the shift in participants' understanding from a linear perspective to a more complex, interconnected understanding of the factors influencing child healthy weights. Participants used an array of systems thinking concepts to describe child healthy weight promotion. In pre-interviews, participants most often named drivers of child healthy weights (e.g., access to affordable, nutritious food; sleep; physical activity) but not how those drivers were associated with health outcomes. For example, one participant mentioned that "they [children] need to eat more fruits and vegetables and get enough sleep to stay generally healthy," without elaborating on how these factors interact with each other or with broader health outcomes (participant 31).

By post-interviews, participants acknowledged the interconnectedness of those drivers more often, emphasizing that child health promotion is generally characterized by relationships and emergent behaviors. For example, one participant noted, "Improving access to healthy food isn't enough on its own; we need to also consider how physical activity, sleep, and things like stress interact to impact a child's overall health down the line" (participant 23). Further, some participants identified how changes to these drivers could have ripple effects on other drivers downstream, noting that "it's getting a community garden started, and suddenly more people are involved, and it grows into this bigger thing...it's not always planned, but you can see how the little actions connect and create this positive vibe in our community from building community and educating people about local resources and building trust" (participant 11).

While participants used cause-and-effect thinking to describe relationships among the drivers of child healthy weights in both the pre- and post-interviews, there was a notable shift to discussing healthy child weights with increasing complexity by the post-interviews; one

participant noted that "Yeah, no, it's not simple. . .there are so many layers and levels [to promoting healthy child weights]" (participant 8). Finally, by the post-interviews, the visual tools category emerged, wherein participants often referred to the need for models, maps, graphs, and diagrams to visualize the structure and dynamics of child healthy weights in their community. Often, discussions of using visual tools corresponded with discussions about complex system literacy. For instance, one participant mentioned that "The CLD helped so much but it also was like, okay, we need more graphs and diagrams in our work for funding and just like for educating others to understand what complex systems are" (participant 9).

Quantitatively, the largest pre- to post-interview code frequency increase in the systems thinking concepts category was the "recognizing interconnections: links, components, or pieces" code (from 17 to 36 codes). The second and third highest pre- to post-interview code frequency increase in the systems thinking concepts category was the "transformation" (from 3 to 16 codes) and the "recognizing relationships" (from 8 to 18 codes) codes, respectively. The largest pre- to post-interview code frequency decrease was "collaboration" (from 84 to 71 codes). There was no to little change in several codes, including "delays" (from 4 to 5 codes) and "diffusion" (from 2 to 4 codes). Example quotes of these codes can be found in the S2 Table.

## Evolving system insights

The main theme identified in this category is "Enhanced Systems Understanding". This theme captures the improvement in participants' comprehension of the dynamics of systems and their components from pre- to post-interviews. In pre-interviews, participants often acknowledged that childhood obesity is part of a system, noting that "like, child health is part of a larger system and we need to focus on that next in our work" (participant 40). By post-interviews, participants shifted to acknowledging system components (e.g., family needs, structural racism, community resources) and how they are related nonlinearly. One participant noted that "It's like when kids engage in sports and there's like this initial positive impact on their activity levels, and then you expect a decrease in obesity rates but it's not like that; like increasing intensity might not lead to a decrease in obesity. . .impact sometimes levels out and you realize that addressing children's health is more complex. . .you can't rely on cause-and-effect" (participant 17).

While participants saw increases in their surface and mid system insights, they also saw smaller increases in their deep system insights. For example, some participants started anticipating the potential consequences of coalition actions on system behavior, noting that "When we talked about adding parks or healthier food options, I always thought about the bigger picture. It's not just about playgrounds or lunches; it's about how these changes could ripple through our community and shape how families live and how healthy our kids grow up to be" (participant 42).

Fig 1 displays the pre- to post-interview code frequency changes for the system insights category. Nearly all aspects of system insights showed improvement. Notably, "acknowledgement of components of a system" increased the most, indicating a heightened awareness of the individual elements that contribute to system functioning. Similarly, some mid-level system insights, such as "what is generic structure" and "nonlinear relationships," displayed gains, demonstrating a deeper grasp of the fundamental patterns and dynamics within systems. Furthermore, "where to intervene" maintained a consistently high level, suggesting that participants recognized the strategic points for intervention and change within systems. The only exceptions to this upward trend were "static snapshot" and "linear thinking," which remained relatively stable or decreased slightly.

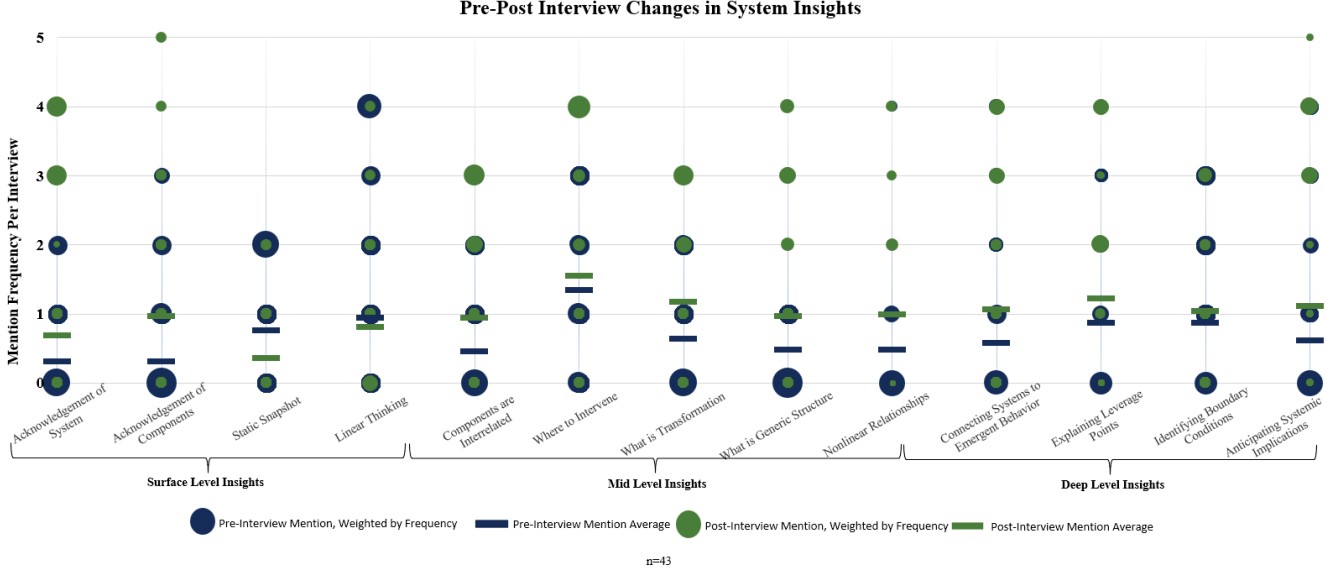

**Fig 1. Pre-post interview changes in system insights.** Mention frequency refers to the number of times a code was mentioned within each interview. Blue dots represent *pre*-interview code mentions per interview, weighted by frequency (the larger the dot, the more it was mentioned across interviews at that frequency). Blue dashes represent the average code mention frequency across pre-intervention interviews. Green dots represent *post*-interview code mentions per interview, weighted by frequency (the larger the dot, the more it was mentioned across interviews at that frequency). Green dashes represent the average code mention frequency across post-intervention interviews.

## Health equity thinking and action

The main theme identified is in this category was "Advancing Health Equity Awareness and Action." This theme captures the notable progress in participants' understanding and commitment to addressing health equity, as reflected in their increased discussions on social and structural determinants of health, and their discussions about the kinds of strategies, programs, or interventions needed to promote child healthy weights. In pre-interviews, participants often referred to health disparities in their health equity thinkings and building community capacity and increasing health options when describing health equity actions. For instance, one participant mentioned that "...they [children] have different outcomes in their health and so they often face more challenges staying healthy" (participant 35); while another noted that "Everyone needs the tools to address health disparities...we need additional trainings like in cultural competency...and we've already built so many great relationships" (participant 24).

By post-interviews, participants shifted to discussing the social and structural determinants of health. For instance, one participant mentioned that social determinants are "So like not just medical care; it's everything around you affecting your well-being like where you were raised and your income" (participant 15). Further, by post-interviews, participants described specific interventions to reduce deterrents to health behaviors. For example, one participant described targeted actions, noting that "...yeah we talked about identifying and addressing barriers to healthy behaviors. It made me reflect on how we can reduce obstacles for families, like improving access to safe places for physical activity and ensuring there are affordable options for healthier food" (participant 14). Notably, participants shifted to describing social and economic resources more often, noting that "We have been taking action a lot more for

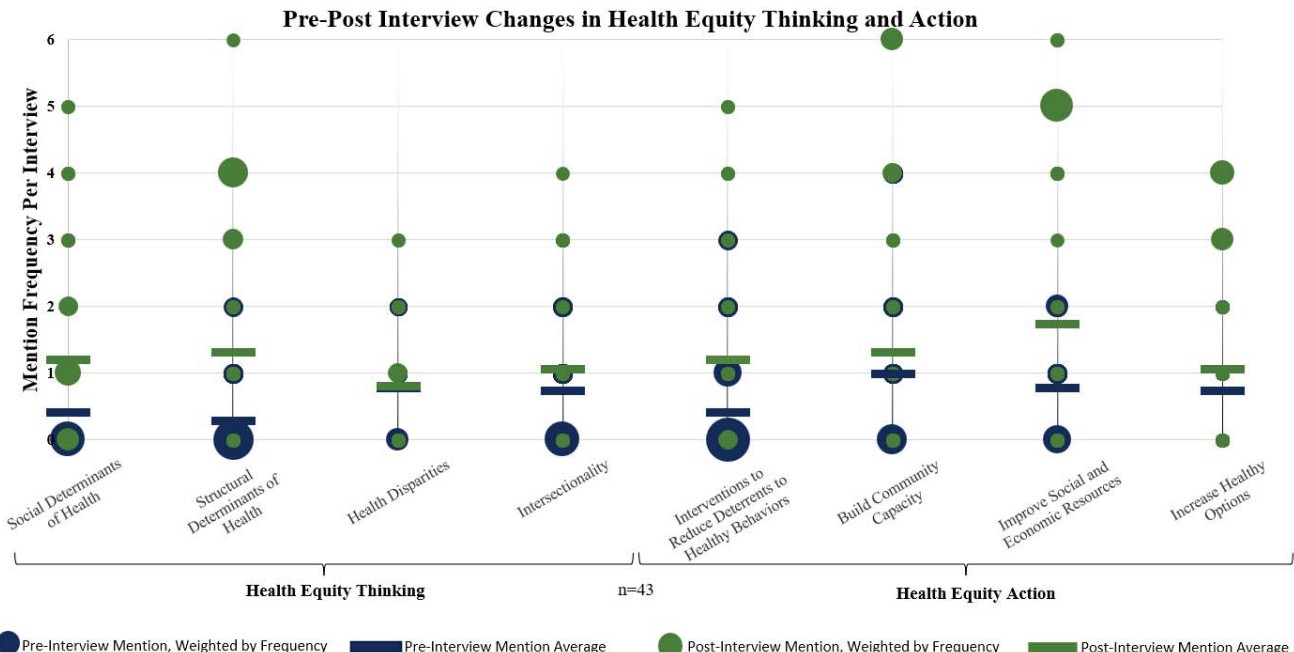

**Fig 2. Pre-post interview changes in systems thinking and action.** Mention frequency refers to the number of times a code was mentioned within each interview. Blue dots represent *pre*-interview code mentions per interview, weighted by frequency (the larger the dot, the more it was mentioned across interviews at that frequency). Blue dashes represent the average code mention frequency across pre-intervention interviews. Green dots represent *post*-interview code mentions per interview, weighted by frequency (the larger the dot, the more it was mentioned across interviews at that frequency). Green dashes represent the average code mention frequency across post-intervention interviews.

sustaining our efforts over a longer period of time. . .I think this really means addressing social and economic factors like unemployment" (participant 16).

As seen in Fig 2, there was a substantial increase in the discussion of "social determinants of health" and an even more pronounced increase in attention toward "structural determinants of health." This signifies a growing acknowledgment among participants of the underlying societal factors and systems that profoundly influence child health outcomes. However, the level of discussion regarding "health disparities" remained relatively consistent, indicating that while participants were attentive to the concept, their emphasis on discussing disparities did not significantly change. Furthermore, there was a marked increase in discussions concerning "specific interventions to reduce deterrents to health behaviors" and "improving social and economic resources." These findings suggest an emerging commitment to implementing targeted actions aimed at mitigating barriers to health and enhancing resources. Conversely, there was less emphasis on "building on community capacity" and "increasing healthy options."

## System insights and health equity

The final theme that emerged was "Integrating System Insights with Health Equity Thinking and Action." This theme highlights the evolving understanding and application of system insights in relation to health equity considerations, reflecting both the complexities and the strategic approaches participants adopted over time.

Fig 3 shows pre-interview code frequency changes, Fig 4 shows post-interview code frequency changes, and Fig 5 shows pre-post interview code frequency changes, each at the intersection of system insights, health equity thinking, and health equity action. Table 2 highlights

**Fig 3. Heatmap of pre-interview changes in system insights, health equity thinking, & healthy equity action.** Code intersection frequency changes fall along a spectrum of dark red (decreased the most, including no change) to dark green (increased the most).

| | | Pre-Interview (n=43) | | | | | | | | | | | | |
|---|---|---|---|---|---|---|---|---|---|---|---|---|---|---|
| | | Surface System Insights | | | | Mid System Insights | | | | | Deep System Insights | | | |
| | | Acknowledge of System | Acknowledgement of Components | Static Snapshot | Linear Thinking | Components are Interrelated | Where to Intervene | What is Transformation | What is Generic Structure | Nonlinear Relationships | Connecting Systems to Emergent Behavior | Explaining Leverage Points | Identifying Boundary Conditions | Anticipating Systems Implications |
| Health Equity Thinking | Social Determinants of Health | 4 | 3 | 10 | 4 | 5 | 4 | 2 | 7 | 4 | 9 | 0 | 0 | 5 |
| | Structural Determinants of Health | 1 | 2 | 7 | 2 | 2 | 1 | 1 | 6 | 2 | 11 | 1 | 1 | 6 |
| | Health Disparities | 3 | 0 | 3 | 2 | 2 | 1 | 6 | 2 | 2 | 5 | 0 | 1 | 1 |
| | Intersectionality | 1 | 0 | 1 | 0 | 2 | 4 | 0 | 1 | 0 | 2 | 0 | 2 | 2 |
| Health Equity Action | Interventions to Reduce Deterrents to Healthy Behaviors | 2 | 2 | 3 | 2 | 6 | 1 | 6 | 6 | 2 | 5 | 5 | 1 | 1 |
| | Build on Community Capacity | 4 | 1 | 1 | 1 | 3 | 1 | 5 | 2 | 2 | 2 | 1 | 0 | 2 |
| | Improve Social and Economic Resources | 4 | 4 | 1 | 5 | 5 | 4 | 11 | 5 | 5 | 4 | 4 | 0 | 3 |
| | Increase Healthy Options | 5 | 8 | 3 | 6 | 2 | 1 | 16 | 5 | 2 | 11 | 6 | 4 | 4 |

other quotes that correspond with Fig 5's heatmap. Overall, there were mixed (increases and decreases) results between surface system insights and health equity thinking and health equity action. For example, acknowledgement that there is a system corresponded with a decrease in discussing the "social determinants of health" and "increase in healthy options", with larger increases in discussing the "structural determinants of health" and "interventions to reduce deterrents to healthy behaviors". One participant highlighted one aspect of this shift by stating,

| | | Post-Interview (n=43) | | | | | | | | | | | | |
|---|---|---|---|---|---|---|---|---|---|---|---|---|---|---|
| | | Surface System Insights | | | | Mid System Insights | | | | | Deep System Insights | | | |
| | | Acknowledge of System | Acknowledgement of Components | Static Snapshot | Linear Thinking | Components are Interrelated | Where to Intervene | What is Transformation | What is Generic Structure | Nonlinear Relationships | Connecting Systems to Emergent Behavior | Explaining Leverage Points | Identifying Boundary Conditions | Anticipating Systems Implications |
| Health Equity Thinking | Social Determinants of Health | 3 | 5 | 2 | 4 | 5 | 12 | 12 | 9 | 8 | 21 | 6 | 3 | 11 |
| | Structural Determinants of Health | 4 | 5 | 2 | 1 | 3 | 8 | 9 | 8 | 4 | 22 | 5 | 8 | 7 |
| | Health Disparities | 3 | 6 | 5 | 1 | 4 | 3 | 8 | 4 | 4 | 7 | 2 | 2 | 1 |
| | Intersectionality | 3 | 3 | 2 | 4 | 3 | 4 | 2 | 1 | 0 | 3 | 0 | 6 | 2 |
| Health Equity Action | Interventions to Reduce Deterrents to Healthy Behaviors | 6 | 8 | 3 | 1 | 7 | 17 | 16 | 8 | 13 | 14 | 13 | 13 | 10 |
| | Build on Community Capacity | 4 | 4 | 6 | 1 | 8 | 9 | 9 | 5 | 7 | 2 | 5 | 2 | 17 |
| | Improve Social and Economic Resources | 5 | 8 | 1 | 4 | 10 | 9 | 17 | 5 | 11 | 12 | 13 | 8 | 6 |
| | Increase Healthy Options | 2 | 8 | 3 | 1 | 6 | 14 | 12 | 2 | 4 | 10 | 8 | 6 | 7 |

**Fig 4. Heatmap of post-interview changes in system insights, health equity thinking, & healthy equity action.** Code intersection frequency changes fall along a spectrum of dark red (decreased the most, including no change) to dark green (increased the most).

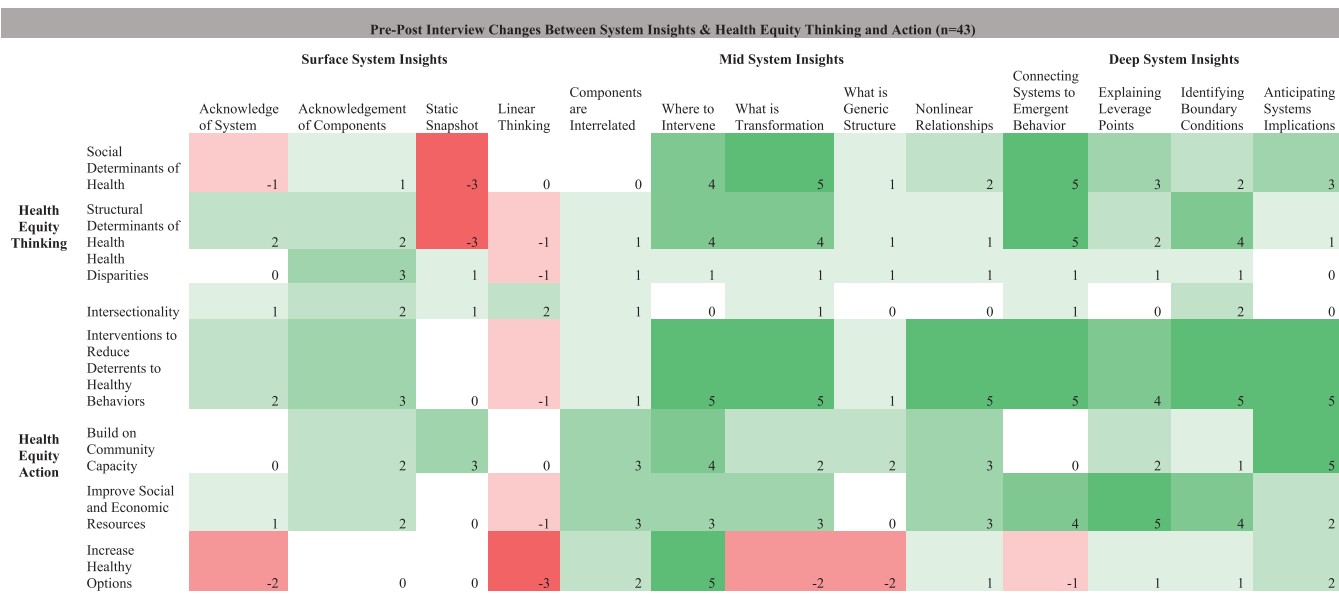

**Fig 5. Heatmap of pre-post interview changes in system insights, health equity thinking, & healthy equity action.** Code intersection frequency changes fall along a spectrum of dark red (decreased the most) to white (no change) to dark green (increased the most). An algorithm was used to color each cell based on the magnitude of change from pre- to post-interview to help guide result interpretation (e.g., -3 indicates a larger decrease from pre- to post-interview than -2).

"Yeah, at first we focused a lot on providing healthy food options and talking about like access to parks. But over time, we realized that we also needed to address deeper structural issues, like improving local policies and reducing systemic barriers to healthy living" (participant 33).

Supporting quotes were chosen to complement Fig 5 and based on their richness in describing code overlaps in the main categories of system insights and health equity thinking and action.

The "static snapshot" and "linear thinking" codes saw the largest decreases, reflecting a move away from simplistic, one-dimensional views of health issues towards a more dynamic and interconnected perspective of childhood obesity prevention. The decrease in "linear thinking" intersected with the concept of "increasing healthy options," suggesting a shift from straightforward solutions to more holistic, systemic approaches. One participant illustrated this shift by saying, "so many of us thought that just adding more healthy food options would help solve the problem. But now we are looking at how all the different factors, like education, community support, and local policies, work together to influence health outcomes" (participant 3).

"Acknowledgement of components" of a system corresponded with the largest increases when discussing "health disparities" and "interventions to reduce deterrents to healthy behaviors". One participant expressed this by stating, "Seeing all the different parts of the system, like healthcare access, schools, and job opportunities, made us realize how they all play a role in perpetuating health disparities. It helped us come up with interventions that really tackle these issues and support healthier habits" (participant 15).

Furthermore, the largest increases in mid-level system insights were seen at intersections such as "where to intervene" with both "interventions to reduce deterrents to healthy behaviors" and "increase healthy options." This shift underscores participants' growing strategic focus on identifying leverage points for effective intervention. As one participant put it, "Knowing where to intervene in the system, like focusing on reducing barriers to physical activity and doing community gardens, can make a bigger impact than just implementing

**Table 2. Supporting quotes at the intersection of system insights and health equity thinking and action.**

| | Supporting Quotes at The Intersection of System Insights and Health Equity Thinking and Action (n = 43) | | |
| --- | --- | --- | --- |
| | **Surface System Insights** | **Mid System Insights** | **Deep System Insights** |
| **Health Equity Thinking** | "Because of it [group model building], we've seen all the different parts of the system, like healthcare access, schools, and income levels, and I think it helped me get why some children deal with more health disparities than others." (participant 17) | "Understanding how different parts of the system work together made me realize that transforming child health means not just talking about eating better but also creating safe places for kids to play and reducing stress at home." (participant 23) | "Understanding how everything in the system is connected made me see that if we focus on building community centers and programs and other capacities, it won't just help kids get more active—it'll bring our families together and strengthen support networks, which can have a huge impact on reducing childhood obesity." (participant 42) |
| **Health Equity Action** | "Once I saw our system, it hit me that fixing up sidewalks and parks isn't just about places for kids to play or exercise and that it's a part of a bigger picture that makes healthy choices easier for everyone." (participant 2) | "Figuring out where to make changes in the system by using the causal loop diagram I think showed us that improving local food stores is like making those healthy choices more accessible and changing the way the whole community eats and not just kids." (participant 21) | "We've [the committee] thought a lot about key leverage points within the system, like boosting job opportunities and affordable housing, and how that can really change the game but also how that may create unanticipated outcomes too." (participant 11) |

isolated health programs like we, as a group, have considered before" participant 1). Overall, by post-interviews, participants no longer emphasized transformative projects focused on increasing healthy eating options as often, noting that intervention and transformation should focus more on reducing social and structural deterrents to healthy behaviors.

Finally, deep system insights, such as "connecting systems to emergent behavior," saw the largest increases when linked with "social determinants of health," "structural determinants of health," and "interventions to reduce deterrents to healthy behaviors." One participant put it simply, "Like all these pieces, like social and structural issues, really fit together and affect each other in complex or like sometimes unintentional ways. And we need to focus on big-picture solutions that tackle these problems head-on and like remove the barriers to promoting child health equity" (participant 19).

## Discussion

This study described changes in systems thinking concepts and system insights among a group of Catalyzing Communities (CC) intervention participants and linked those changes to health equity thinking and action. The current study adds to the growing evidence that links community-based system dynamics approaches to increasing the use of systems thinking concepts in the context of community health promotion, and, to date, is the first evaluation to link systems thinking concepts to health equity thinking and action in context of a whole-of-community intervention that aims to promote child health equity. The following sections review our findings in context of previous CC intervention findings and of the broader literature on systems thinking, health equity, and childhood obesity prevention.

### Systems thinking concepts & insights

The most significant changes in systems thinking concepts observed in participants before and after the CC intervention indicate a transformative shift in their understanding of child obesity. Initially focusing on drivers like access to nutritious food and sleep without considering their associations with health outcomes, participants, post-intervention, demonstrated an understanding of child obesity as a complex adaptive system. Their description of concepts such as "diffusion," "interdependence," and "emergence" aligns with classic characteristics of complex adaptive systems [46]. Additionally, participants emphasized the importance of visual tools, reflecting an awareness of using models, maps, graphs, and diagrams to better visualize

the structure and dynamics of child healthy weights. This finding reflects participants' awareness of using visual tools to understand a complex topic like obesity trends over time, and is consistent with research that discusses increasing the use of systems thinking concepts through the construction of participatory models [47].

The most significant changes in system insights reveal that participants improved their understanding of mid-level and deep-level system insights, including nonlinear relationships and anticipating system behavior. Some participants showcased comprehension of fundamental system patterns and dynamics, consistently maintaining a high level of understanding regarding strategic points for intervention and change within systems. This enhanced system insight can improve decision-making and intervention design in childhood obesity prevention, helping identify key drivers, understand unintended consequences, and design adaptable interventions [32].

Changes seen in participants' use of systems thinking concepts and in levels of systems insights align with the goals and content of the CC intervention activities, which prioritizes small and all-group activities that focus on identifying and linking the determinants and consequences of child healthy weights to potential interventions. For instance, after determinants related to the reference mode are elicited, the CC intervention uses connection circles to visually link the determinants loosely based on their relationship with each other and to begin identifying feedback loops these relationships form within a system. As participants visually represent their mental models of the system that drives child healthy weight trends, they discuss the determinants, feedback loops, and the potential leverage points within the system, promoting the use of systems concepts and systems insights in the process.

In a previous study, we documented how applying community-based system dynamics to design and implement activities that promote insights into the systems driving childhood obesity prevalence helped participants prioritize food insecurity, building power among historically marginalized voices within the community, and support advocacy efforts to promote community-wide change [24]. The current study provides supporting evidence for how prioritization of new actions to influence child healthy weights might be occurring by increasing the use of system concepts and by the evolution of deepening systems insights over time.

Our approach also facilitated the identification of participants' perceptions regarding key determinants influencing child obesity prevalence in their local communities. Participants initially identified drivers such as access to affordable, nutritious food, sleep, and physical activity, but their understanding evolved to recognize the interconnectedness of these factors and their broader implications. Recent research using systems approaches, like community-based system dynamics, has shown that increasing system insights can uncover often overlooked drivers of child healthy weights that have greater potential in which to intervene [8, 12, 24]. Uncovering new drivers of a problem opens possibilities for new, potentially more effective interventions.

Our results reinforce the broader literature indicating that increasing system insights helps participants facilitate improved communication around and anticipate potential unintended consequences of prioritized actions [30, 31, 48, 49]. This is crucial as the behavior of complex systems can be difficult to predict. Recognizing these potential consequences allows us to take steps to mitigate them [50]. For example, one participant noted, "It's not just about playgrounds or lunches; it's about how these changes ripple through our community" (participant 42). This deeper understanding also allowed them to foresee and plan for potential ripple effects, as another participant mentioned, "We've thought a lot about key leverage points like job opportunities and housing, and how that can create unanticipated outcomes too" (participant 11). Increasing participant communication around and anticipate potential unintended consequences of prioritized actions could support research that links the enhancement of

system insights to the design of interventions that are more adaptable to change [51, 52]. Given the constant evolution of childhood obesity challenges [48], it is essential to create interventions that can adjust to new circumstances. For example, an intervention relying on partnerships with local businesses may require adaptation if these businesses close or undergo ownership changes. Understanding the dynamic nature of complex systems allows us to design interventions that are more likely to be sustainable over time.

## Health equity thinking & action

While coding for system concepts and systems insights, health equity thinking and health equity action emerged out of discussions about barriers to achieving healthy child weights. By post-intervention, except for a small increase in code frequencies for health disparities, participants discussed health equity thinking concepts and health equity actions more often in each category. Notably, discussions around "social determinants of health" and "structural determinants of health"for health equity thinking, and "interventions to reduce deterrents to healthy behaviors"and "improve social and economic resources"for health equity action, increased the most. This study is the first to document how the CC intervention may be linked to participant increases in discussing health equity related to child healthy weights.

While we anticipated that increases in system thinking concepts and insights might be associated with elevated health equity thinking and action, given the established association between systems thinking and child health equity concepts and achievement [26, 30], we were surprised by the widespread increases across all codes in these two categories. We hypothesized that the use of systems thinking concepts is connected to discussions involving health equity thinking and action. To explore this, we cross-tabulated these discussions, analyzing the system insights influencing changes in how participants discuss child health equity (described in more detail below). Future research should delve into the content and dosage of GMB activities directly associated with increased health equity thinking and action. Considering the relatively smaller increase in code frequency related to discussing health disparities, there may be an opportunity to incorporate GMB activities in the CC intervention that explicitly address issues surrounding disparities in local child health outcomes and access to safe, affordable, nutritious foods among different child and family populations.

## Systems thinking & health equity intersections

One goal of the CC intervention is to ensure that understandings and proposed solutions that emanate from the use of systems thinking concepts are (1) sensitive and competent from an equity perspective; (2) designed to avoid the chance that disparities may widen as an unintended consequence; and (3) accelerate progress in those least well-served by current approaches. Thus, one metric of success for the CC intervention is the use of systems thinking concepts to prioritize actions aimed at reducing health disparities within child populations. When examining the intersection of systems insights, health equity thinking, and health equity action in the code frequency heat map, participants tended to use mid and deep system insights to describe social and structural determinants of health. Similarly, participants used surface system insights less frequently than mid or deep system insights to describe increasing healthy options for children. These patterns may point to how specific aspects of health equity thinking and action, including reducing child health disparities, requires deeper system insights.

Narrowing in on specific changes between levels of system insights and healthy equity thinking and action, participants decreased their use of linear thinking (code category: surface system insight) to describe each category of health equity action. This pattern may be linked to

participants using deeper systems insights to describe interventions and combinations of interventions that incorporate considerations related to social disadvantage and social and structural determinants of health. Indeed, participants increased in their use of mid and deep system insights to describe each category of health equity actions, though to a lesser extent for increasing healthy options. One explanation for the lack of attention to increasing healthy options in post-interview conversations is that most participants already had experience in understanding and implementing strategies to increase healthy options for children and shifted to areas of less familiarity and potentially more complexity.

Finally, participants seem to have begun thinking about intersectionality and health disparities, with most participants using surface system insights to acknowledge the components of intersectionality and linear thinking to describe intersectionality. Because intersectionality has been linked to addressing health, economic, and racial crises [53], it may be important for the CC intervention, and for whole-of-community interventions more broadly, to spend more time cultivating deeper system insights into how intersectionality and health disparities play a role in health equity action.

The Getting to Equity framework is built on creating interventions that synergize across health equity actions, that guide decisions about what interventions or supports might be combined [26]. Our identified systems insights and health equity action interactions suggest that participants were thinking along those lines in large part because of their mid and deep system insights. For example, participants often cited the need to address discrimination (code: reduce deterrents to healthy behaviors, which aligns with the connection between discrimination and deterrents to healthy options in the Getting to Equity framework) and increase housing subsidies and tax credits (code: improve social and economic resources). Because our coding analysis found that these discussions coincided with deeper systems insights codes, we hypothesize that systems insights such as describing nonlinear relationships and explaining leverage points within the system may facilitate prioritization of synergistic health equity actions. This enhanced understanding led to more informed and strategic decision-making. For instance, stakeholders were able to prioritize strategies that addressed root causes of childhood obesity rather than just symptoms. Whole-of-community childhood obesity prevention interventions, and community-based interventions in general, may benefit from activities that have been shown to increase systems thinking, such as GMB [54] and participatory modeling [55].

## Limitations of the study

This study used interview data that was not corroborated with other data sources or perspectives. Participants may have chosen to provide responses that align with perceived societal or researcher expectations, particularly when discussing sensitive topics like health equity. Additionally, we combined interviews from across four communities, which may have influenced our results. Future research could examine how community context influences the use of systems thinking concepts and health equity. Future research could also focus on the design and testing of new survey inventories that will help whole-of-community interventions mechanistically connect systems thinking concepts to health equity.

Furthermore, several limitations specific to our coding process should be noted: (a) the occurrence of a code does not convey the depth or quality of discussion about a concept; for instance, a brief surface-level insight is counted the same as a lengthy, in-depth surface-level insight; (b) code co-occurrences reflect that two different concepts were mentioned in the same paragraph or sentence but do not indicate whether or how they were discussed in relation to each other; (c) the use of systems thinking concepts within an interview does not necessarily translate to an understanding of systems thinking or health equity thinking or actions

consistent with that understanding; and (d) the absence of a code does not mean that the participant does not think or act in a particular way, it simply means that they did not express it during the interview. Acknowledging these limitations underscores the need for future studies to incorporate multiple data sources, validate coding processes, and explore the influence of community context on systems thinking and health equity.

The differences in race, ethnicity, and lived experiences between the interviewers and interviewees, along with the lack of pre-existing relationships, may have influenced the interview responses and findings. Power dynamics and potential cultural misunderstandings could have lead to guarded or socially desirable responses from participants. The interviewers' backgrounds may have affected their interpretation of responses, and participants might have been reluctant to fully disclose their thoughts due to perceived differences. Additionally, the absence of rapport-building prior to the pre-intervention interviews might have impacted the openness and honesty of the responses, despite efforts to inform participants about the research goals and the interviewers' interests.

## Supporting information

**S1 Checklist. COREQ (Consolidated criteria for Reporting Qualitative research) checklist.**
(PDF)

**S1 Table. Generalized Catalyzing Communities meeting sequence and activities across communities.** Adapted from Calancie et al. (2022) [7]. *Hypothesized activities that help build participants' systems thinking. **Evidence share varies by community (e.g., connecting early childhood education and health; promoting community health improvement through more equitable food systems).
(DOCX)

**S2 Table. Categories, codes, definitions, and illustrative quotes from participant interviews.**
(DOCX)

**S1 File. Interview protocol.**
(DOCX)

## Acknowledgments

We would like to thank the Catalyzing Communities participants for their engagement in this project. We would like to acknowledge the important feedback from Shiriki Kumanyika that helped guide the interpretation of results and the discussion. We would also like to acknowledge the important contributions of Cora A. Kakalec and Muyang Wu, students who helped code and analyze the qualitative data included in this study.

## Author Contributions

**Conceptualization:** Travis R. Moore, Larissa Calancie, Erin Hennessy, Christina D. Economos.

**Data curation:** Travis R. Moore.

**Formal analysis:** Travis R. Moore, Larissa Calancie.

**Funding acquisition:** Christina D. Economos.

**Methodology:** Travis R. Moore, Julie Appel, Christina D. Economos.

**Supervision:** Travis R. Moore.

**Validation:** Julie Appel, Christina D. Economos.

**Visualization:** Travis R. Moore.

**Writing – original draft:** Travis R. Moore, Larissa Calancie, Julie Appel, Christina D. Economos.

**Writing – review & editing:** Travis R. Moore, Larissa Calancie, Erin Hennessy, Julie Appel, Christina D. Economos.

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
