## [Decision Letter · Decision Letter 0]

3 Jun 2024

PONE-D-24-05161Changes in systems thinking and health equity considerations across four communities participating in Catalyzing CommunitiesPLOS ONE

Dear Dr. Moore,

Thank you for submitting your manuscript to PLOS ONE. After careful consideration, we feel that it has merit but does not fully meet PLOS ONE’s publication criteria as it currently stands. Therefore, we invite you to submit a revised version of the manuscript that addresses the points raised during the review process.

The manuscript is on a research topic of high interest and within the scope of PLOS One.

The reviewers have made very thorough and constructive analyses and suggestions with which I fully agree, and which will be key to improving the manuscript quality.

In my view:

- some concepts should be clarified (e.g. systems thinking), the background should provide more information on system dynamics and on measuring effectiveness, the description  of methods should be clear, the quantitative and qualitative results should be clearly portrayed, the discussion of results should take into account the approaches in use (e.g. interviews), and the timelines of all components of the study should be shown in a clearer format.

- the authors should make sure that all the relevant information is made available so that the study can be replicated.

Accordingly, if the address all referee comments, the manuscript has the potential to make a relevant contribution to PLOS One.

We look forward to receiving your revised manuscript.

Kind regards,

Monica Duarte Correia de Oliveira

Academic Editor

PLOS ONE

 [C.E. received funding that supported this study from the JPB Foundation (PR0580). The JPB Foundation did not play a role in the study design, data collection and analysis, decision to publish, or the preparation of the manuscript. https://www.jpbfoundation.org ].  

Reviewers' comments:

Reviewer's Responses to Questions

**Comments to the Author**

1. Is the manuscript technically sound, and do the data support the conclusions?

Reviewer #1: Partly

Reviewer #2: Partly

2. Has the statistical analysis been performed appropriately and rigorously? 

Reviewer #1: N/A

Reviewer #2: N/A

3. Have the authors made all data underlying the findings in their manuscript fully available?

Reviewer #1: No

Reviewer #2: No

4. Is the manuscript presented in an intelligible fashion and written in standard English?

Reviewer #1: Yes

Reviewer #2: Yes

5. Review Comments to the Author

Reviewer #1: The authors an important and persistent problem in public health. The paper needs to be improved by addressing the following issues:

General comment: Clarify the meaning of Systems Thinking and this is used in the research. Clarify what do you mean by the Systems Thinking skills of the stakeholders and project participants. Are these related to the adoption of a formal Systems approaches to tackle child obesity such as the Whole Systems Approach to Diet and Health Weight and/or the general "Thinking in Systems" in healthcare even if a non formal systems approach is being implemented (all health contexts and problems can be conceptualised from a Systems thinking perspective even if no formal Systems based intervention or approach is implemented).

General comment: Clarify how System Dynamics (SD) is being used in this research. There is a great deal of confusion on how Systems Dynamics is deployed in this research and to what purpose. Which SD tools and methods were used in the research and how these informed the data collection and data analysis methods in this research. There is a great amalgam of concepts included in the research (systems thinking, system dynamics, health equity, child obesity, catalizing communities interventions) and these need careful structuring and unpicking so that it is clear how each concept is used and how they are linked in a coherent way to inform the research.

1. Place the paper clearly in the literature on Child Obesity and System Dynamics as there has been a wealth of literature on these topics. Indicate how your research differs from and complements past research in this topic highlighting the novelty and contribution of your paper.

2.Page lines 61 to 70. Indicate how did you select the 4 evaluation questions. Clarify the role of "health equity" in the research as this was not developed prior to the evaluation questions.

3. Provide background information about the project you are gathering the data from.

4.Page 5. Background. Systems Thinking. Clarify the definitions of Systems Thinking and System Dynamics as these are different, albeit linked concepts. Provide a brief literature overview of the applications of System Dynamics in obesity control.

5.Page 6. Background. Provide an overview of the System Dynamics literature on Health Equity and how these relate and inform the Health Equity framework used in this study.

6. Page 12. Interview protocol. Provide references for the work done by the scholars, which informed your protocol.

7. Data Analysis. Provide more details om how the GMB activities described in Table 1 were used to help the community teams in the 4 locations develop their Systems Thinking skills. Were there any differences observed between the 4 locations.

8.Provide details about the health outcomes of the CC interventions in the 4 sites. Were there any improvements in terms of child obesity and health equity outcomes. If so, how this is linked to the development of the Systems Thinking skills by the stakeholders in the 4 sites.

9. Discussion: Discuss the findings of the research with respect to past academic literature especially with regard to the influence of GMB activities on group learning and improvements in shared problem understanding and policy making.

10. Discussion: Discuss the findings with regard to the measurement of stakeholders Systems Thinking skills and how this influences decision making quality. How these relate to the success of the interventions and health and equity outcomes.

Reviewer #2: With respect to the PLOS ONE criteria:

1. This study presents the results of primary scientific research.

2. To the best of my knowledge the results have not been published elsewhere.

3. I have some concerns/suggestions regarding the technical standard and sufficiency of the description of the analyses, which I detail below.

4. Whilst in the most part the conclusions are presented in an appropriate fashion and are supported by the data, there are a couple of areas where I do not feel that this is the case (see below).

5. The article is presented in an intelligible fashion and is written in standard English, although I do have some specific comments on how this could be improved (see below).

6. To the best of my knowledge, this research appears to meet all applicable standards for the ethics of experimentation and research integrity. However, I would ask the authors to consider whether the two student coders meet the conditions for authorship, and if so then they should be authors not just mentioned in the acknowledgements.

7. This article does not adhere to appropriate reporting guidelines and community standards for data availability. Specifically, I would suggest that the authors consider using the COREQ checklist for reporting qualitative research (or SRQR). If possible, it would be nice for the anonymised interview data to be made available, but I recognise that this may not be possible under the consent / research ethics agreement.

Summary

The authors present an interview based analysis of changes in the systems thinking, health equity thinking and health equity actions of participants involved in a community intervention for addressing childhood obesity (‘Catalyzing Communities’). They examine the intersections between systems insights and health equity thinking and action, and how these change pre/post engagement in Catalyzing Communities.

Major comments

- The quantitative approach is rather limited in the sense that: (a) the occurrence of a code does not convey the depth/quality about which someone discussed a concept, e.g. a brief surface-level mention would ‘count’ as much a lengthy in-depth and thoughtful discussion about a concept (b) code cooccurrences reflect that two different concepts were mentioned in an interview, but tells us nothing about whether/how they were discussed in relation to each other (c) the use of systems thinking language within an interview does not necessarily translate to a person understanding the concepts of systems thinking or acting in a way consistent with an understanding of systems thinking (d) an absence of a code does not mean that the participant doesn’t think/act in a particular way, it simply means that they didn’t say as much in an interview. I would expect to see these limitations noted in the manuscript at a minimum. Better still would be to include richer data in the form of quotes (see below).

- The authors state that they have conducted a mixed methods analysis, but the results are solely related to the quantitative analysis (frequencies of codes pre/post). These show some trends but are somewhat limited (see above) and complementary, richer insights could come from the qualitative aspect of the analysis, i.e. by drawing out themes and illustrating these with quotes. There are large sections of text in the results that list the cooccurrences of code categories that are shown already in the heatmap – instead, the authors could share more insightful findings through quotes that illustrate how systems insights were influencing health equity thinking/action. The authors could consider an additional table of quotes alongside the heatmaps that illustrate how different systems insights overlap with health equity thinking/action. Also - what did the participants say directly about how systems thinking had influenced their thinking and action about health equity? It seems odd not to include an analysis of that in addition to the code cooccurrence frequencies.

- The authors should include the topic guide (including any differences between the pre/post intervention interviews), either in the main manuscript or as supplementary material. When presenting frequencies of codes, i.e. the number of mentions, it is important for the reader to understand what was asked since this will naturally influence whether or not a participant will talk about something and how they talk about it. It would also be instructive for the authors to reflect on their positionality as interviewers and how this may have affected the interview responses and the findings.

- The study compares interviews conducted pre and post the first year of the CC. Over this time period many things will have influenced how the participants think and talk about healthy equity, and how they think and talk about systems (including the first interview itself, the possibility of which should be noted). Therefore increases in code frequencies alone cannot shed much light on whether the two are linked (and certainly cannot be used to attribute increases in health equity thinking to an increase in systems insights). However, through deeper qualitative analysis and illustrative quotes it might be possible for the authors to build up a richer understanding of how these things are related and influenced each other.

- The heat map encapsulates two things at the same time: the cooccurrence of systems insights with health equity thinking/action AND the change in this pre/post intervention. It would be clearer to de-couple these two things by providing an additional 2 tables that show the numbers of cooccurrences pre and post intervention separately. The authors could then describe the correlations in both cases before highlighting key changes. The key changes would also then be placed within the context of the absolute numbers, e.g. is a change in occurrence of +3 pre/post intervention due to a rise from 0 to 3, or from 12 to 15 for example?

- Some of the discussion on page 21 (lines 431-447) does not seem firmly rooted in the results. In particular, it needs to be clearer how the following statements are backed up by the results:

--- ‘The approach used in our study not only revealed participants' perceptions on key determinants influencing child obesity in local communities but also emphasized the importance of increasing system insights.’

--- ‘Firstly, our approach facilitated the identification of participants' perceptions regarding key determinants influencing child obesity prevalence in their local communities.’

--- ‘Secondly, our results reinforce the broader literature indicating that increasing system insights helps participants anticipate potential unintended consequences of prioritized actions.’

--- ‘Thirdly, the enhancement of system insights aids in designing interventions that are more adaptable to change.’

Additionally, please make it clearer how the following statement in the Abstract is backed up by the results: ‘Our findings reveal significant shifts in systems thinking… leading to improved systems insights and strategic intervention recognition’.

Minor comments

- It is not clear whether ‘health equity action’ means participants talked about possible health equity actions, or actual actions taken. The reader would benefit from a clearer definition of this term and how it is used in the paper.

- Page 7-8: ‘The whole-of-community intervention’ section would fit better in the background, not the methods, as it is about an intervention described elsewhere not the methods used within this study.

- Table 1 would be better as supplementary material rather than feature in the main body of the text since it is about an intervention reported elsewhere. There is also no reference to Table 1 in the main body of the text.

- I find the use of the term ‘systems thinking’ in this manuscript confusing. Usually, and indeed in several places in the paper, ‘systems thinking’ refers to an approach/method. Yet the authors also refer to ‘shifts in participants’ systems thinking’, by which I think they mean changes in their use of systems thinking concepts when talking about childhood obesity in the interviews (not a change in their use of systems thinking methods in practice or a change in the methodology itself). It would be helpful to find a different term to ‘systems thinking’ when referring to participants’ use of systems thinking concepts within interviews.

- More detail is required in the methods, for example were the interviews audio recorded, transcribed, anonymised? Was consent taken? What was the sampling strategy (was every committee member asked?) Did you reach saturation? Using the COREQ/SRQR will help to guide the sorts of information required in the methods section.

- In the text it says that Table 2 shows the 43 pre/post match participants, but it actually shows the coalition-committee members. Please revise Table 2 to include the details of the subset of committee members that were interviewed. Additionally, in Table 2 please clarify that it is ‘Community Race and ethnicity (%)’ and also include race and ethnicity as one of the committee characteristics. Please spell out the ‘NH’ abbreviation.

- I would suggest that the Table in the supplementary material goes instead in the main manuscript (moving the current Table 2 to the supplementary material would make room for it). I would also find it helpful to have 2-3 illustrative quotes for each code within this Table.

- I note that two students coded the transcripts - did a more senior researcher code a sample of the transcripts?

- In the Results, the ‘Systems thinking concepts’ section needs illustrative quotes (or a table or figure) so that the reader is able to get closer to the data not just a high level summary.

- Page 19 ‘the first process evaluation to …’ - is this study a ‘process evaluation’? This is not how the authors describe it elsewhere in the paper.

6. PLOS authors have the option to publish the peer review history of their article (what does this mean?). If published, this will include your full peer review and any attached files.

Reviewer #1: No

Reviewer #2: No

---

## [Author Response · Author response to Decision Letter 0]

23 Jun 2024

Please see the Response to Reviewers document for a properly formatted version of this text.

Response to Reviewers

We appreciate the insightful and constructive feedback provided on our manuscript, "Changes in Systems Thinking and Health Equity Considerations Across Four Communities Participating in Catalyzing Communities." The editor’s and reviewers’ thorough review has been invaluable in enhancing the clarity and rigor of our work. We have carefully considered each of your comments and suggestions, and we are pleased to submit our revised manuscript along with detailed responses to your feedback.

In the following sections, we address (not in bold) each point raised by the reviewers (in bold), outlining the modifications made to the manuscript and providing justifications for our approaches. We believe these revisions have significantly strengthened our study, and we hope our responses satisfactorily address all concerns.

Editor Comments

1. Some concepts should be clarified (e.g. systems thinking), the background should provide more information on system dynamics and on measuring effectiveness, the description of methods should be clear, the quantitative and qualitative results should be clearly portrayed, the discussion of results should consider the approaches in use (e.g. interviews), and the timelines of all components of the study should be shown in a clearer format.

a. Thank you for summarizing the thoughtful feedback from reviewers one and two. As described below, we have attempted to address each of these concerns. We provide a summary of how we have addressed each one here:

i. Clarify concepts such as systems thinking: we have addressed this by reworking the background section on systems thinking. We provided a clearer definition, some background on its origin, and placed it in context of the broader literature.

ii. Provide more information on system dynamics and on measuring effectiveness: System dynamics is outside the scope of this manuscript. We have removed mention of system dynamics in the manuscript. Instead, we define and provide additional detail about community-based system dynamics, which was informed by system dynamics but not as relevant to the purpose of the paper, which is to evaluate changes in systems thinking, insights, and health equity thinking and action. 

iii. The description of the methods should be clear: We have now included the COREQ checklist to ensure that we are systematic about including important information about the design and methods of the qualitative study. 

iv. The quantitative and qualitative results should be clearly portrayed: we addressed this by rewriting most of the results section, creating additional figures and tables, and by integrating participant quotes throughout. 

v. The discussion of results should consider the approaches in use (e.g. interviews): We have integrated a discussion of the approaches used in both design and analytical methods in the limitations section. 

vi. The timelines of all components of the study should be shown in a clearer format: we have provided clear dates for when each component of the study occurred. 

2. The authors should make sure that all the relevant information is made available so that the study can be replicated.

a. We provide the study’s full codebook with illustrative quotes for each code, the interview protocol/guide, and the COREQ checklist of items that should be included in reports of qualitative research. 

b. Additionally, the redacted data supporting this study can now be found on OSF open science database: DOI 10.17605/OSF.IO/VCDYG.

Reviewer 1 Comments

Major Comments

1. Clarify the meaning of Systems Thinking and this is used in the research. Clarify what do you mean by the Systems Thinking skills of the stakeholders and project participants. Are these related to the adoption of a formal Systems approaches to tackle child obesity such as the Whole Systems Approach to Diet and Health Weight and/or the general "Thinking in Systems" in healthcare even if a non formal systems approach is being implemented (all health contexts and problems can be conceptualized from a Systems thinking perspective even if no formal Systems based intervention or approach is implemented).

a. We appreciate this comment and agree that systems thinking requires further clarification, definition, and a clear link to its operationalization. In response, we have revised the "Systems Thinking" section in the background. This revision includes a more precise definition of systems thinking in the context of our project, a brief overview of systems thinking based on the literature, an explanation of the connection between systems thinking and system insights, and a description of the literature that informed the development of our codebook.

b. Revision Location: Pages 5-6, lines

2. Clarify how System Dynamics (SD) is being used in this research. There is a great deal of confusion on how Systems Dynamics is deployed in this research and to what purpose. Which SD tools and methods were used in the research and how these informed the data collection and data analysis methods in this research. There is a great amalgam of concepts included in the research (systems thinking, system dynamics, health equity, child obesity, catalyzing communities interventions) and these need careful structuring and unpicking so that it is clear how each concept is used and how they are linked in a coherent way to inform the research.

a. We agree that mentioning both System Dynamics and Community-based System Dynamics within the same manuscript can be confusing without additional clarification. System Dynamics is mentioned in the context of Community-based System Dynamics, an approach developed and promoted by Peter Hovmand that integrates systems thinking and participatory methods to engage communities in understanding and addressing complex issues. Therefore, while System Dynamics as a field is not directly relevant to this paper, Community-based System Dynamics is highly relevant, as it is a core element of the CC intervention.

To resolve this confusion, we have removed mentions of System Dynamics from the paper, including the discussion, and refer only to Community-based System Dynamics. Additionally, we have defined Community-based System Dynamics in the introduction to clarify its meaning. We also defined group model building, a tool of Community-based System Dynamics, in the introduction. When group model building is described in detail in the methods section (now moved to the background section per reviewer 2’s suggestion), we ensured that System Dynamics is not mentioned, further clarifying that it is a tool used within Community-based System Dynamics. Finally, we place health equity in context of the literature that uses Community-based System Dynamics.

b. Revision Locations: Page 3 (Introduction), Page 7 (Health Equity Thinking and Action), Page 11 (The Whole-of-Community Intervention)

3. Place the paper clearly in the literature on Child Obesity and System Dynamics as there has been a wealth of literature on these topics. Indicate how your research differs from and complements past research in this topic highlighting the novelty and contribution of your paper.

a. We agree that our study could be better situated within the childhood obesity prevention literature. To address this concern, we have positioned the CC intervention within the broader context of childhood obesity prevention literature, specifically focusing on the use and promotion of systems thinking and group model building, a tool used in Community-based System Dynamics (page 7). Since System Dynamics is not a key part of our paper, we have removed mentions of it throughout the text. However, we have clearly situated the paper within the literature on Community-based System Dynamics. This context is provided in the discussion section, where we relate our findings to our own and others' previous research using Community-based System Dynamics. This revised discussion can be found on page 27.

b. Revision Locations: Page 7, Page 9/Whole-of-Community Interventions, and Page 27

Minor Comments

4. Page lines 61 to 70. Indicate how did you select the 4 evaluation questions. Clarify the role of "health equity" in the research as this was not developed prior to the evaluation questions.

a. We clarified the process the research team used to select the four evaluation questions and the role of health equity in the research (Page 4). We explained that health equity played a crucial role by examining how participants' understanding and actions related to health equity evolved throughout the intervention. Although the concept of health equity was not explicitly developed before formulating the evaluation questions, it became an integral part of the research focus. The evaluation questions were designed to capture changes in participants' health equity thinking and actions, emphasizing the importance of addressing health disparities alongside systems thinking.

b. Revision Location: Page 4, Lines 63-80

5. Provide background information about the project you are gathering the data from.

a. We believe that it is very important to provide background information about the project that we are gathering data from. This information can be found in the background section, on page 10, where we also renamed the section to “The Whole-of-Community Intervention: Catalyzing Communities.

b. Revisions Location: Page 10

6. Page 5. Background. Systems Thinking. Clarify the definitions of Systems Thinking and System Dynamics as these are different, albeit linked concepts. Provide a brief literature overview of the applications of System Dynamics in obesity control.

a. See reviewer 1’s #1 comment above for more information on how we addressed this important topic. Further, we added text in the Whole-of-Community Interventions section to situate the paper in the literature that discusses the use of community-based system dynamics, and the use of group model building, in childhood obesity prevention interventions. 

7. Page 6. Background. Provide an overview of the System Dynamics literature on Health Equity and how these relate and inform the Health Equity framework used in this study.

a. We provide more information about how we addressed this cocnern in reviewer 1’s #3 comment. We situated health equity within the broader literature on interventions that use community-based system dynamics and group model building to promote health equity. We also connected these studies to the value of the current study, which more explicitly connects systems thinking to health equity thinking and action. Finally, we added a new paragraph that explains how and why the framework was used in the current study. 

b. Revision Locations: Page 7, Page 9/Whole-of-Community Interventions, and Page 27

8. Page 12. Interview protocol. Provide references for the work done by the scholars, which informed your protocol.

a. While the references were provided in the second paragraph of the interview protocol section, we also provided the references upfront when the scholars are named, in the first paragraph of the interview protocol section. We hope that this makes it more evident that we are referencing the work done by other scholars.

b. Revision Location: Page 13

9. Data Analysis. Provide more details on how the GMB activities described in Table 1 were used to help the community teams in the 4 locations develop their Systems Thinking skills. Were there any differences observed between the 4 locations.

a. We agree that there could be a description linking how group model building activities described in table 1 were used to develop systems thinking in intervention participants. To address this comment, we developed a new paragraph in the “The Whole-of-Community Intervention” section, which states and substantiates that the activities are used to help participants identify key patterns and trends, understand the components and interrelationships within the system, visualize changes over time, and see how variables are interconnected. We also added an asterisk to the activities in Table 1 that we believe are connected to building systems thinking. Additionally, we discuss how key GMB activities may be linked to increasing systems thinking in the discussion (e.g., in the section on systems thinking concepts and insights). Table 1 has also now been moved to supplemental material to conserve space.

b. Revision Location: Page 11

10. Provide details about the health outcomes of the CC interventions in the 4 sites. Were there any improvements in terms of child obesity and health equity outcomes. If so, how this is linked to the development of the Systems Thinking skills by the stakeholders in the 4 sites.

a. We would be delighted to provide more details about the health outcomes of the CC intervention at the four sites, and we do have manuscripts in development that address this specifically. However, in this manuscript, our focus is on linking systems thinking to health equity thinking and actions rather than health equity impact. We believe this upstream approach is crucial for promoting health equity in the long-term, as evidenced by the broader literature connecting upstream determinants of health to downstream health outcomes. Naturally, the time horizon for realizing these downstream health outcomes is quite long. We look forward to reporting on the intervention's impact in a forthcoming manuscript.

b. Revision Location: NA

11. Discussion: Discuss the findings of the research with respect to past academic literature especially with regard to the influence of GMB activities on group learning and improvements in shared problem understanding and policy making.

a. We agree with this comment and have added additional references and discussion in the "Systems Thinking Concepts and Insights" subsection of the discussion section. Notably, we included a reference that links GMB to improved problem understanding, stating:

"The approach used in our study not only revealed participants' perceptions of key determinants influencing child obesity in local communities but also emphasized the importance of increasing system insights. Our approach facilitated the identification of participants' perceptions regarding key determinants influencing child obesity prevalence in their local communities. Recent research using systems approaches, such as Community-based System Dynamics, has shown that increasing system insights can uncover often overlooked drivers of child healthy weights that have greater potential for intervention (8, 12, 25). Uncovering new drivers of a problem opens possibilities for new, potentially more effective interventions. Additionally, our results reinforce the broader literature indicating that increasing system insights helps participants improve communication and anticipate potential unintended consequences of prioritized actions (31, 32, 52, 53). This is crucial, as the behavior of complex systems can be difficult to predict. Recognizing these potential consequences allows us to take steps to mitigate them (54). Furthermore, the enhancement of system insights aids in designing interventions that are more adaptable to change (55, 56). Given the constant evolution of childhood obesity challenges (52), it is essential to create interventions that can adjust to new circumstances. For example, an intervention relying on partnerships with local businesses may require adaptation if these businesses close or undergo ownership changes. Understanding the dynamic nature of complex systems allows us to design interventions that are more likely to be sustainable over time."

b. Revision Location: Page 27

12. Discussion: Discuss the findings with regard to the measurement of stakeholders Systems Thinking skills and how this influences decision making quality. How these relate to the success of the interventions and health and equity outcomes.

a. We appreciate the thoughtfulness of this comment. Because we did not measure decision-making quality or link quality to the success of the intervention and health equity outcomes, we cannot substantively address this request. However, we did revise the final paragraph of the discu

---

## [Decision Letter · Decision Letter 1]

15 Aug 2024

Changes in systems thinking and health equity considerations across four communities participating in Catalyzing Communities

PONE-D-24-05161R1

Dear Dr. Moore,

We’re pleased to inform you that your manuscript has been judged scientifically suitable for publication and will be formally accepted for publication once it meets all outstanding technical requirements.

Kind regards,

Monica Duarte Correia de Oliveira

Academic Editor

PLOS ONE

Additional Editor Comments (optional):

In my view the authors made a serious revision, addressing mine and the referees' comments, and the manuscript is now worth publication by PLOS One. Congratulations to the authors.

I ask the authors to verify that all data involved in the study is made available to readers.

Reviewers' comments:

Reviewer's Responses to Questions

**Comments to the Author**

1. If the authors have adequately addressed your comments raised in a previous round of review and you feel that this manuscript is now acceptable for publication, you may indicate that here to bypass the “Comments to the Author” section, enter your conflict of interest statement in the “Confidential to Editor” section, and submit your "Accept" recommendation.

Reviewer #1: All comments have been addressed

Reviewer #2: All comments have been addressed

2. Is the manuscript technically sound, and do the data support the conclusions?

Reviewer #1: Yes

Reviewer #2: Yes

3. Has the statistical analysis been performed appropriately and rigorously? 

Reviewer #1: N/A

Reviewer #2: N/A

4. Have the authors made all data underlying the findings in their manuscript fully available?

Reviewer #1: No

Reviewer #2: (No Response)

5. Is the manuscript presented in an intelligible fashion and written in standard English?

Reviewer #1: Yes

Reviewer #2: Yes

6. Review Comments to the Author

Reviewer #1: The study is interesting and relevant in the current public health context of child obesity. The replies to the comments are satisfactory and the required corrections have been made to the paper.

Reviewer #2: (No Response)

7. PLOS authors have the option to publish the peer review history of their article (what does this mean?). If published, this will include your full peer review and any attached files.

Reviewer #1: No

Reviewer #2: No

---

## [Editor Report · Acceptance letter]

21 Aug 2024

PONE-D-24-05161R1 

PLOS ONE

Dear Dr. Moore, 

I'm pleased to inform you that your manuscript has been deemed suitable for publication in PLOS ONE. Congratulations! Your manuscript is now being handed over to our production team.

Kind regards, 

on behalf of

Professor Monica Duarte Correia de Oliveira 

Academic Editor

PLOS ONE